# Physical properties of the HIV-1 capsid from all-atom molecular dynamics simulations

Juan R. Perilla[1,2] & Klaus Schulten[1]

Human immunodeficiency virus type 1 (HIV-1) infection is highly dependent on its capsid. The capsid is a large container, made of ~1,300 proteins with altogether 4 million atoms. Although the capsid proteins are all identical, they nevertheless arrange themselves into a largely asymmetric structure made of hexamers and pentamers. The large number of degrees of freedom and lack of symmetry pose a challenge to studying the chemical details of the HIV capsid. Simulations of over 64 million atoms for over 1 μs allow us to conduct a comprehensive study of the chemical–physical properties of an empty HIV-1 capsid, including its electrostatics, vibrational and acoustic properties, and the effects of solvent (ions and water) on the capsid. The simulations reveal critical details about the capsid with implications to biological function.

[1] Department of Physics and Beckman Institute, University of Illinois at Urbana-Champaign, Urbana, Illinois 61801, USA. [2] Department of Chemistry and Biochemistry, University of Delaware, Newark, Delaware 19716, USA. Correspondence and requests for materials should be addressed to J.R.P. (email: jperilla@udel.edu).

The family of retroviruses is characterized by their ability to incorporate viral DNA into a host cell's genome[1,2]. Most retroviruses infect cells during mitosis when the chromatin is exposed to the cytoplasm[1,3]. Conversely, the genus of lentiviruses, like the human immunodeficiency virus type 1 (HIV-1), have evolved to infect nondividing cells. Since the host cell's chromatin is protected by the nucleus, the HIV-1 replication process requires coordination between reverse transcription of viral RNA and nuclear import[3]. Viral RNA is encased in a shell made of the capsid protein CA[4–7]. Originally thought to play a trivial role in the infection process, it is now well established that the viral capsid fulfills several essential functions[1]. In particular, capsid involvement in the prevention of innate sensor triggering[8–11], regulation of reverse transcription[3,12] and regulation of the nuclear import pathway[1,13–15] is of central importance to the successful infection of a host cell[3].

Structurally, CA consists of a single monomer composed of two independent domains, the C-terminal domain (CTD) and the N-terminal domain (Fig. 1a)[16,17]. In solution, CA readily forms dimers connected by their CTDs[16]. At higher concentrations of CA and at high salt concentrations ($>1$ M NaCl), the protein assembles into elongated tubes[5]. Such tubes are made of the hexameric form of CA, arranged in a honeycomb lattice[5,17–21]. Assembled HIV-1 capsids are highly asymmetric cone-shaped structures[5,22], with sizes ranging from 100 to 200 nm long and 45 to 50 nm wide (Fig. 1b)[1,5,22], that are made of pentamers and hexamers following Eberhard's theorem[21]. The location of the pentamers induces high curvature and permits the closure of the capsid[4,5,21]. Indeed, in the HIV-1 capsid seven pentamers are located at the base and five pentamers at the tip (Fig. 1b).

The function of a CA monomer is to assemble into a complete capsid, while the function of the assembled capsid is to infect a living cell; the chemical–physical properties of the assembled capsid are essential to elucidating its biological function[23–25]. In the present study, the investigation of the chemical–physical properties of the HIV-1 capsid without genome require the simulation of an entire virus-like particle (VLP), namely a 1.2 µs, 64 million atom molecular dynamics (MD) simulation (Fig. 1b)[5,17,26]. Analyses of the trajectory reveal atomic details as well as global emergent properties of an entire HIV-1 VLP. In particular, the simulations reveal spontaneous translocation of water and ions through the capsid core, a molecular process related to reverse transcription[27]. Furthermore, analysis of the electrostatic potential produced by the empty capsid reveals electrostatic equivalence between the interior and exterior of the capsid. In addition, analysis of the distribution of ions around the capsid reveals a binding pattern and specific binding sites on the capsid for chloride and sodium. Finally, the mechanical vibrations and normal modes of the empty capsid reveal possible pathways related to capsid uncoating. Altogether, our findings set the stage for a better understanding—from an atomic-resolution perspective—of the emergent properties of the HIV-1 capsid and their relationships to the infective cycle.

## Results

**Stability of the HIV-1 capsid.** We began our analysis by assessing the stability of the HIV-1 capsid model, formed by 1,300 copies of the CA, shown in Fig. 1. Stability of the individual constituents of the capsid, namely CA pentamers and hexamers, was evaluated by means of root mean squared deviation (r.m.s.d.) from the starting structure over the course of the simulation (Fig. 2a). The r.m.s.d. values were calculated excluding parts of the protein known to be disordered or mobile[28], such as the cyclophilin-A (cypA) binding loop and residues 210 to 220 directly after α-helix 11. The average r.m.s.d. for CA hexamers

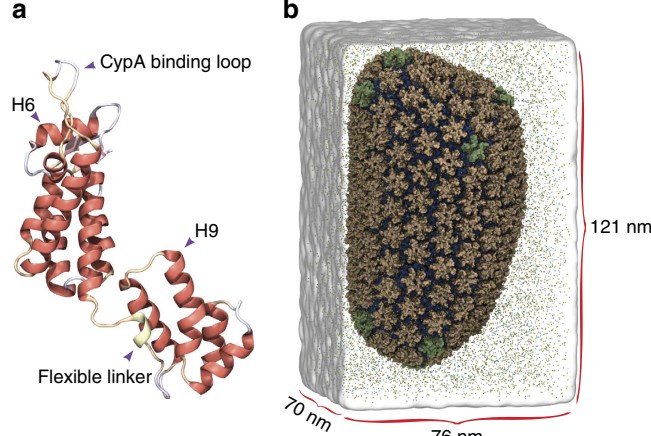

**Figure 1 | The HIV-1 capsid. (a)** The HIV-1 capsid is made of a single capsid protein (CA), containing 11 α-helices and a $3_{10}$ helix. **(b)** CA arranges into a fullerenic cone, consisting of pentamers (green) and hexamers (tan). The fully solvated HIV-1 capsid model without genome, including neutralizing ions and 150 mM NaCl, contains a total of 64,423,983 atoms[5].

and pentamers of $3.5 \pm 1.1$ Å shows the local stability of the capsid. Furthermore, the measured r.m.s.d. over the trajectory for CA pentamers and hexamers of $2.7 \pm 0.6$ and $3.8 \pm 1.0$ Å, respectively, shows that the pentamers are on average more rigid than the hexamers. Similarly, the variability within the ensemble of hexamers in the capsid was $3.7 \pm 0.9$ Å as compared with $2.6 \pm 0.5$ Å for the 12 pentamers. In addition, the stability of the capsid as a whole was evaluated by calculating the r.m.s.d. matrix between pairwise structures over the course of the trajectory (Supplementary Fig. 1 and Supplementary Movie 1); the distance matrix shows that after 400 ns all capsid structures are within $3.5 \pm 1.0$ Å of each other.

Changes in the cross-sectional area and height of the capsid are related to its global stability (Fig. 2b–d). Therefore, both the height and cross-sectional area were calculated along the three principal axes of inertia of the capsid (Fig. 2b). Interestingly, during the first 400 ns of simulation, a shrinking of the capsid was observed, evidenced by a reduction in both the height and cross-sectional area at a rate of 0.025 nm ns$^{-1}$ and 0.113 nm$^2$ ns$^{-1}$, respectively (Fig. 2b,c). After 400 ns, the capsid height and cross-section reach a plateau for the remaining 800 ns of the simulation that is used in the subsequent analyses.

Examination of the water density in the interior and exterior of the capsid reveals the origin of capsid shrinkage (Supplementary Fig. 2a). During the first 100 ns of simulation, a decay in water density was observed inside the capsid, followed by stabilization around $1.000 \pm 0.002$ gm cm$^{-3}$. Importantly, the rates of water moving in and out of the capsid are $20.1 \times 10^3 \pm 1.6 \times 10^3$ and $19.7 \times 10^3 \pm 1.5 \times 10^3$ water molecules per ns, respectively (Supplementary Fig. 2b). The transfer rates of water indicate thermal equilibrium between the interior and exterior of the capsid and imply that the HIV-1 capsid is capable of replacing all of its contained water in $\sim 1$ µs. Remarkably, the transfer rates observed for HIV-1 are in stark contrast with order of magnitude smaller transfer rates observed for poliovirus capsids[29].

**Electrostatics and ion permeability of the VLP.** The capsid protein is negatively charged; in fact, at pH 7, the net electric charge of the whole capsid is $-3,528$ e. The electrostratic potential due to the distribution of electric charge of a fully

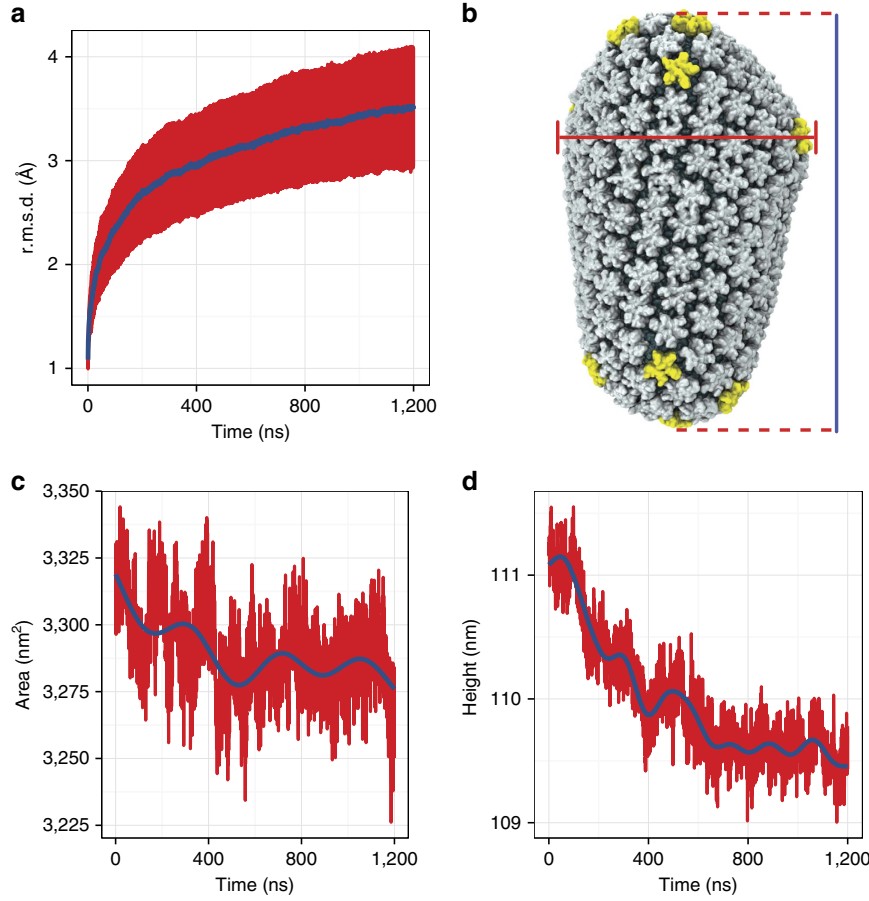

**Figure 2 | Stability of the HIV-1 capsid.** (**a**) Time evolution of the root mean squared deviation (r.m.s.d.) for hexamers and pentamers, and the moving average with a window size of 10 ns is shown in blue. (**b**) For all area and height calculations, the three principal moments of inertia of the entire capsid define the $x$, $y$ and $z$ axes. The cross-sectional area is estimated as the area of an ellipse where the major and minor axes are the maximal distance between parts of the capsid along the $\hat{x}$ and $\hat{y}$ axes. The height of the capsid is defined as the longest distance from the tip (bottom) to the base (top) along the $\hat{z}$ axis. (**c**) Time evolution of the capsid's cross-section, and the moving average with a 10 ns window size is shown in blue. (**d**) Time evolution of the height of the HIV-1 capsid; the moving average with a 10 ns window size is shown in blue.

solvated capsid, that is, explicitly including water and ions, was calculated at 20 ps intervals over the course of the simulation, over a grid containing $350 \times 353 \times 600$ bins along the $\hat{x}$, $\hat{y}$ and $\hat{z}$ direction, respectively. The resulting electrostatics maps were averaged over the last 400 ns of simulation. Snapshots of the electrostatic maps at regular intervals over the trajectory, averaged over 10 ns of simulation, are presented in Supplementary Fig. 3. The time-averaged electrostatic potential is shown in Fig. 3a,d. Notably, a gradient of up to 7 V is observed between the CypA binding loop and the inner core of CA (Fig. 3c); in addition, the innermost layer of the VLP constitutes an isopotential volume spanning the entire capsid (Fig. 3b). Intriguingly, the effective electrostatic potential of hexamers and pentamers is remarkably similar. Furthermore, the regions of the solvent inside and outside of the capsid are at the same electrostatic potential as shown in Fig. 3b.

The presence of water and ions in the simulation box offers a unique view into the interactions between the capsid and its native environment. Taking advantage of the small variations within each of the ensembles of hexamers and pentamers, 186 hexamers/12 pentamers were r.m.s. fitted to a common reference; then, occupancy of chloride and sodium ions was measured over the last 400 ns of simulation, resulting in a total sampling per hexamer or pentamer of $186 \times 0.4\,\mu s = 74.4\,\mu s$ and $12 \times 0.4\,\mu s = 4.8\,\mu s$, respectively. The analysis reveals regions of high occupancy for chloride (cyan) and sodium (yellow) (Fig. 4). Comparison of our results with published crystal structures show that our observation of chloride at the centre of the hexameric rings is accurate[20]. However, in our simulations, the binding of chloride to other regions is also detected, in particular assisted by highly conserved residues K70 and K180 (ref. 30). In addition, we also observe the presence of sodium near the surface of the capsid at multiple sites, especially near genetically important residues[30] E71, E75, E76, E79, E212 and E213.

Binding of ions to the whole capsid reveals an interesting overall pattern (Fig. 4): Chloride ions form an inner layer adjacent to the surface of the capsid, while sodium binds to the exterior of the capsid. Examination of the transfer rates for both ionic species present in the simulation reveal that ions are in thermal equilibrium between the interior and exterior of the capsid (Supplementary Fig. 2c,d). Interestingly, over the course of the simulation, we were able to observe numerous translocation events of chloride through the central pores in hexamers and pentamers (Supplementary Movie 2), and sodium through the pores located between adjacent CTD dimers (Supplementary Movie 3). Notably, the inwards and outwards transfer rates for sodium are $9.4 \pm 2.4$ and $8.5 \pm 1.1$ molecules per ns, respectively. In contrast, we observed inwards and outwards transfer rates twofold that of sodium for chloride, $22.2 \pm 2.9$ and $20.8 \pm 3.3$ molecules per ns, respectively.

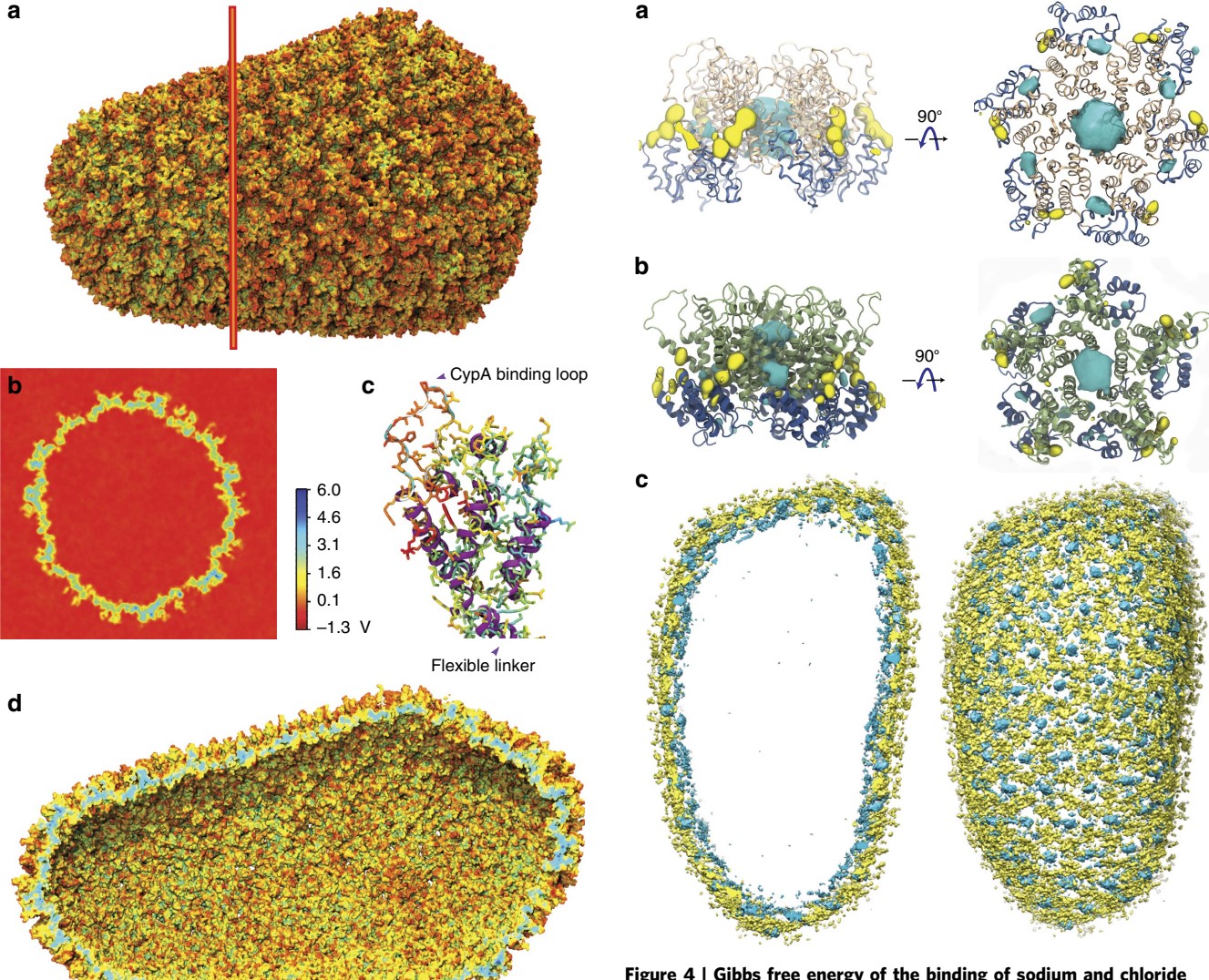

**Figure 3 | Electrostatic potential averaged over the final 400 ns of the HIV-1 capsid simulation.** The electrostatic calculation includes all capsid atoms and all solvent molecules for a total of 64,423,983 atoms. The bar scale indicates the magnitude of the electrostatic potential in Volt, ranging from −1.3 V (red) to 6.0 V (blue). (**a**) Exterior view of the HIV-1 electrostatic potential. The red line indicates the location of the cross-section shown in **b**. (**b**) Cross-section of the electrostatic potential of the HIV-1 capsid. The bulk in the interior and exterior of the capsid assume the same electrostatic potential values, namely −1.3 V. (**c**) Electrostatics of the N-terminal domain of CA. The cypA binding loop and α-helix 4 (Fig. 1a) show a significant potential difference to the inner core of the capsid in **d**.

**Figure 4 | Gibbs free energy of the binding of sodium and chloride ions to CA hexamers and pentamers.** Free-energy isosurfaces at $\Delta G = -1.0\,\text{kcal mol}^{-1}$. Sodium is represented in yellow and chloride in blue. (**a**) Binding of sodium and chloride to CA hexamers. The free energies were calculated over a total sampling time of 74.4 μs. (**b**) Binding of sodium and chloride to CA pentamers. The free energies were calculated over a total sampling time of 4.8 μs. (**c**) Distribution of ions on the entire HIV-1 capsid. A layer of chloride ions can be observed at the interior of the capsid, while sodium binds preferentially to the exterior of the capsid.

At each binding site, ion occupancies are directly related to the binding free energies of both ionic species[31]; therefore, from the occupancy maps, we derived a three-dimensional (3D) potential of mean force (PMF) for both pentamers and hexamers, as explained in Methods and shown in Fig. 4. The resulting PMFs illustrate that irrespective of the oligomeric (hexamer or pentamer) state of CA, the ion binding pattern is remarkably similar. Nevertheless, the PMFs show that binding of chloride in the central pore, near R18, is weaker for hexamers than it is for pentamers, as demonstrated by their Gibbs free energy $\Delta G_{\text{CLA}}^{\text{hex}} = -1.5 \pm 0.9\,\text{kcal mol}^{-1}$ and $\Delta G_{\text{CLA}}^{\text{pent}} = -2.7 \pm 1.1\,\text{kcal mol}^{-1}$, respectively. The difference in binding energies between pentamers and hexamers suggests that

chloride ions translocate more easily through hexameric channels than they do through pentameric channels. Similarly, sodium binding at the hinge region (residues 144 to 148)[16] and the threefold interface between hexamers exhibits a lower affinity, $\Delta G_{\text{SOD}}^{\text{hex}} = -1.8 \pm 0.5\,\text{kcal mol}^{-1}$ compared with chloride in any other binding site. Remarkably, the free energies of binding for chloride and sodium ions are similar to the experimentally measured value for binding of sodium to wild-type thrombin ($-2.3\,\text{kcal mol}^{-1}$)[32]; nonetheless, we find lower values comparing the free energy of binding of sodium to glutamate receptors ($-10.0 \pm 5.0\,\text{kcal mol}^{-1}$)[33], and the sodium symporter ($-10.0 \pm 2.0\,\text{kcal mol}^{-1}$)[34]. Overall, the distribution of ions and exchange rates reveal that the HIV-1 capsid is able to translocate ions with remarkable specificity; such specificity may play an important role during infection, as it would assist and filter translocation of molecules that are key for successful infection (for example, DNA nucleotides[27]).

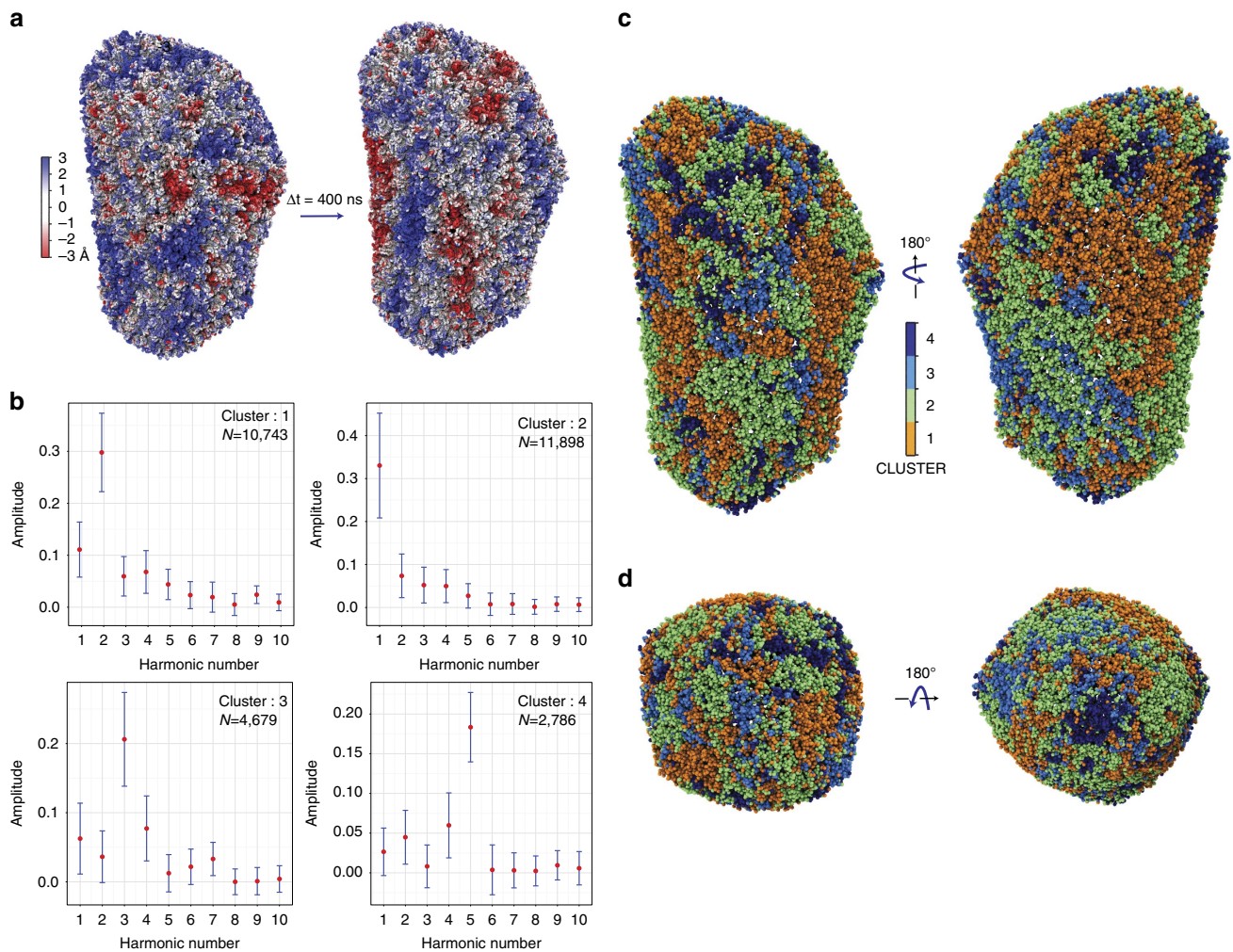

**Figure 5 | Acoustic properties of the HIV-1 capsid during the last 400 ns of simulations.** (**a**) Structural fluctuations of the capsid observed between two states separated 400 ns apart, ranging from $-3$ to $+3$ Å (red to blue). (**b**) Periodograms of the time series of the capsid motions. Four classes of periodograms were found: the two largest classes ($N = 10,743$ and $N = 11,898$) are dominated by the two lower fundamental frequencies (2.38 and 4.76 MHz); conversely, the two smallest classes ($N = 4,679$ and $N = 2,786$) are dominated by two higher fundamental frequencies (7.14 and 11.9 MHz). The medoids of each class are represented in red dots in **b**. The s.d. within each class is represented in blue error bars. (**c**) Projection of the periodogram clusters onto the structure of the capsid indicating the location of each cluster. (**d**) Top (base) and bottom (tip) view of the capsid coloured by the periodogram clusters. The four clusters are coloured orange, green, cyan and dark blue, respectively.

**Acoustic properties of the HIV-1 capsid.** Structural fluctuations of the capsid during the simulation revealed complex dynamics. Deviations from the equilibrium position of individual $C_\alpha$s were projected over the capsid for different time points, as described in Methods (Fig. 5a and Supplementary Movie 4). The projection of the variations reveals an oscillatory behaviour of the surface of the capsid with magnitude $3.0 \pm 1.0$ Å. Interestingly, similar oscillations have been observed for lipid vesicles[35]. These 'surface waves' are spread over the capsid and dynamically correlate large regions of the capsid (Fig. 5a and Supplementary Figs 4 and 5).

The capsid is a closed container that can be characterized based on its acoustic properties. For this purpose, periodograms were calculated for each $C_\alpha$ time series (Fig. 5b). A periodogram is the squared magnitude of the discrete Fourier transform and identifies the most dominant frequencies in a time series[36]. Therefore, the periodograms permit one to identify the most dominant frequencies in the measured time series for every $C_\alpha$ in the HIV-1 capsid. Because of the large number of $C_\alpha$s ($>150,000$,) analysing the periodograms by visual inspection is impractical.

Instead, by using each periodogram as a descriptor for every fifth $C_\alpha$ and the Manhattan distance between descriptors as a similarity score between $C_\alpha$s, every fifth $C_\alpha$s were classified into clusters using a partition around medoids (PAM) algorithm[37]. The silhouette criteria (Supplementary Fig. 6)[37] was used to establish the optimal number of clusters, indicating four of them (Fig. 5b). The same clustering analysis was performed for halved and quartered fragments of the simulation that also resulted in four major clusters (Supplementary Fig. 4).

Remarkably, each of the four clusters identified by the PAM analysis are dominated by a different fundamental frequency (Fig. 5b and Supplementary Fig. 4). Indeed, the two largest clusters ($N = 10,743$ and $N = 11,898$) are dominated by the two lower fundamental frequencies (2.38 and 4.76 MHz); conversely, the two smallest clusters ($N = 4,679$ and $N = 2,786$) are dominated by two higher fundamental frequencies (7.14 and 11.9 MHz). The clusters were mapped to their locations on the structure of the capsid (Fig. 5c,d). Interestingly, the highest frequency cluster (four) is located at the tip of the capsid and around the circumference of the base. On the other hand,

lower frequency clusters are located on the large regions spanning between the base and tip of the capsid.

As previously mentioned, analysis of halved and quartered fragments of the trajectory reveals that oscillations of the capsid can still be clustered for shorter lengths of the simulation. Nonetheless, the mapping of the frequencies on to the capsid structure identify different frequency regions for different fragments of the simulation (Supplementary Fig. 5). These differences between frequency-to-structure mappings suggests that the waves we observe are not stationary. We therefore conclude that the waves observed behave like capillary waves common in several types of membranes and fluids[35]. A remaining question, beyond the scope of the present study, is the relationship between capillary waves and permeation of water and ions through the capsid.

**Mechanical modes of the HIV-1 capsid.** Normal modes are informative to elucidate the collective motions of flexible molecules that underlie changes in their conformation[38]. The displacement patterns obtained from normal-mode analysis (NMA) provide insights into the mechanical nature of biological nanomachines and can be used to derive conformational transitions related to biological function[39]. A common practice is to extract slow motions from MD trajectories using principal component analysis (PCA) that reveal principal components (PCs) akin to normal modes[40,41]. In the context of viral capsids, several studies have been performed to extract such collective motions by using a plethora of methods[42–46]. Nonetheless, since all NMA studies of entire viral capsids were performed *in vacuo*[45], a long-standing question in physical virology is the influence of hydrodynamic effects resulting from the presence of solvent in the vibrational modes of viral capsids[45].

The vibrational modes of a fully solvated HIV-1 capsid were obtained in the present study based on all-atom MD simulations. In particular, PCA was employed by performing singular value decomposition of the last 400 ns of the trajectory (described in detail in Methods). The PCs were calculated using the $C_\alpha$ atoms ($N = 150,528$), thus resulting in a total of $d = 3 \times N - 6 = 451,578$ degrees of freedom. Convergence of the subspace spanned by the PCs was evaluated for different numbers of modes and lengths of the trajectory using the mutual similarity measurement[40], as explained in Methods and illustrated in Supplementary Fig. 7. A total of $n = 20,000$ frames were employed for the PCA calculations. The number of PCs that can be extracted from the simulation is given by $\min(d,n) = 20,000$ (ref. 47). Interestingly, 300 modes are required to account for over 80% of the variance observed in the simulations (Fig. 6a); similarly, only 100 modes are neessary to account for over 60% of the variance (Fig. 6a). Projection of the last 400 ns of the simulation onto the first two PCs further demonstrates the stability of the capsid (Supplementary Fig. 8). In addition, for the lowest 800 modes the percentage of the capsid involved in each mode was quantified using the so-called participation number[44]. The participation number reveals that the modes are not localized, and that the PCs involve large portions of the capsid ranging from 70 to 84% of the entire capsid (Fig. 6b).

Projection of the displacement patterns of the first and second PCs (which together contribute 25% of the overall variance) onto the structure of the capsid reveal complex motions of the capsid subunits (Fig. 6c,d and Supplementary Fig. 9, respectively). The motions are represented as vectors in the figures, indicating the displacement of a given atom. The pattern of the motions is intriguing, as a belt can be observed that effectively divides the capsid into two hemispheres, corresponding to the tip and the

base of the cone (Fig. 6d). Additionally, related to the capillary waves observed in the previous section, vortices are observed in the motions throughout the structure of the capsid (Fig. 6c,d). Remarkably, the first 100 PCs reveal the absence of a distinct radial breathing mode, and show that all modes exhibit a combination of displacements that imply motion both radially and within the surface of the capsid. The first 100 PCs are included in Supplementary Dataset 1 that can be visualized using the NormalModeWizard available in visual molecular dynamics (VMD).

## Discussion

The HIV-1 capsid is a finely tuned nanodevice that modulates several molecular events during HIV-1 infective cycle by interacting with multiple cellular host factors[1]. The structural stability of the capsid during different stages of the infective cycle remains poorly understood. In the present study, despite observing a global shrinking of the capsid compared with our original model[5], the overall shape of the capsid and the structure of CA hexamers and pentamers remained largely unaltered over 1.2 μs of simulation time. Such global and local stability further supports the fullerenic model of the HIV-1 capsid, as well as the stability of the capsid interfaces, formed by dimers and hexameric/pentameric rings. Our observation that a shrinkage of the capsid is followed by a stable structure is consistent with coarse-grained simulations of HIV-1-like particle assembly, where a global arrangement of the VLP occurs followed by local re-arrangement of the VLP structure[48,49]. In addition, our results suggest that a closed capsid, irrespective of its cargo, once assembled, remains stable and closed while still permitting the passage of water and ions.

So far, acoustic and normal mode analysis for large macromolecular systems have been limited to symmetric particles or coarse-grained representations. In the present study we determined acoustic properties and PCs from all-atom simulations of a solvated capsid. We found that the capsid exhibits an oscillatory behaviour manifested as surface waves. Our analysis indicates that the fundamental frequencies associated with such waves occur in the ultrasound regime (2.38 to 11.9 MHz). To our surprise, the fundamental frequencies were associated with different regions of the capsid. Interestingly, the tip of the capsid, where pentamers are closer to each other, presents the highest fundamental frequency. In addition, our NMA results suggest that the capsid exhibits long-range collective dynamics. Such correlations over long distances may play important roles during infection, as they permit the capsid to transfer information to/from distant regions of its structure. This allosteric communication across the capsid could serve a function during nuclear import, during which interactions between the nuclear pore complex and the capsid modulate infection[1] (Supplementary Fig. 10).

A significant result from the present study is the discovery of binding of cations and anions in genetically important regions of the assembled capsid. We are not the first group to observe ions bound to CA in high-order assemblies; in addition to water molecules, chloride ions were detected by X-ray crystallography at multiple interfaces of a planar hexameric lattice (including the central pore, the CTD dimer and the threefold symmetry axis)[20]. However, whether chloride or sodium ions play a biological role is unclear, as they may be substituting for other ions available in the cytoplasm. Furthermore, not all of the binding sites we observe have also been observed experimentally; particularly, sodium has not yet been observed to interact with the hexameric lattice in the X-ray structures. Moreover, the observed electrostatic signature of

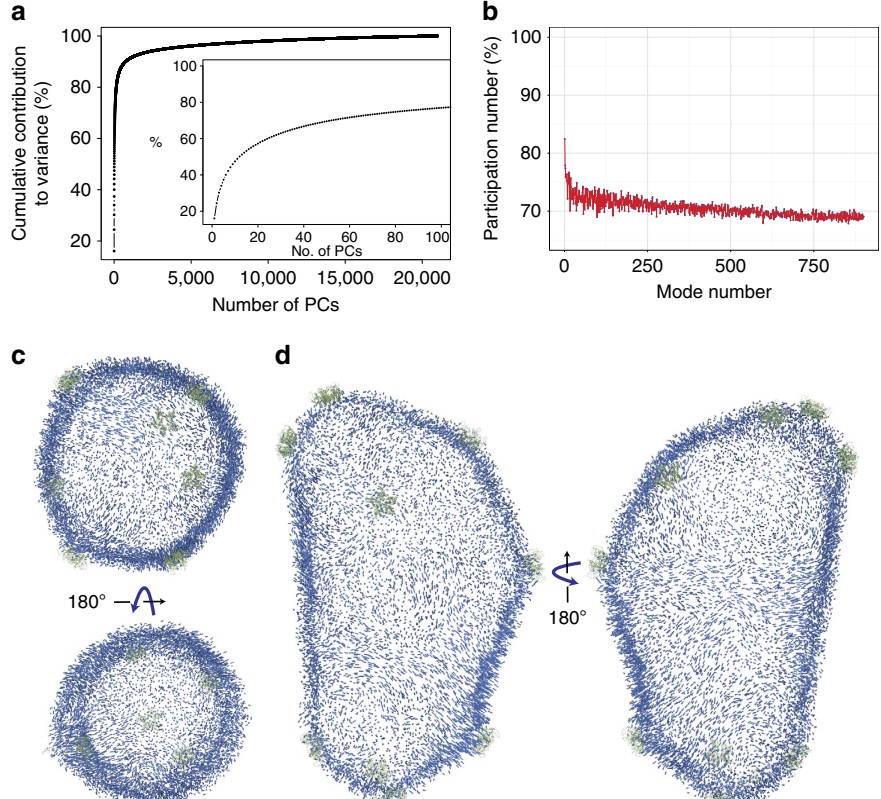

**Figure 6 | Mechanical motions of the HIV-1 capsid.** Collective motions were measured using principal component analysis over the last 400 ns of the capsid trajectory. (**a**) Cumulative contribution to variance from all the principal components. Inset: cumulative contribution to variance from the first 100 principal components. (**b**) Collectiveness of the first 800 principal components according to their participation number[44]. (**c,d**) Projection of the displacement patterns for the first principal component, the direction of the displacement is represented by the direction of the arrow; similarly, the magnitude of the displacement is represented by the length of the arrow. Locations of pentamers are highlighted in green.

the assembled capsid together with the ion binding sites could indicate the existence of protein–protein interfaces for as yet undiscovered cellular factors. Interestingly, the distribution of charge on the surface of the capsid of another retrovirus, the Rous sarcoma virus, is essential for successful infection[50]. Finally, our observation of ion-specific binding sites could also explain the requirement for high-salt concentrations for *in vitro* assembly of CA tubes and VLPs.

Remarkably, we also observe that ions translocate through chloride- and sodium-specific channels within the surface of the capsid. While homeostasis could be a simple explanation for ion translocation, there are other biological functions of the capsid that could require the existence of such channels[3,29,51,52]. For instance, reverse transcription (which is coupled to capsid uncoating) requires that DNA nucleotides be able to move from the exterior to the interior of the capsid[3,16,27]. Therefore, it is possible that the channels found within the capsid serve to translocate small molecules, including nucleotides, necessary during different stages of the infective cycle.

Importantly, in authentic virions, deviations should be expected from the distribution of ions and electrostatics presented here due to the presence of HIV-1 single-stranded RNA and other proteins in the interior of the capsid. In particular, the native HIV-1 capsid contains two copies of its negatively charged ~9.7 kb-long genomic RNA[53] that likely induces osmotic and mechanical pressures between the interior and exterior of the capsid; a similar shift in pressure has also been proposed for poliovirus capsids[29]. Remarkably, changes in pressure of the capsid have been observed during reverse transcription in

time-lapse atomic force microscopy experiments that result in rupture of the capsid at the narrow end[54]. In the context of *in vivo* cargo, the ions channels identified in the present study could help regulate the osmotic shock produced during reverse transcription of the genomic RNA.

The present work is a comprehensive study of the chemical–physical properties of the HIV-1 capsid—an important therapeutic target. Through the combination of state-of-the-art MD simulations with scalable, robust and model-free statistical analysis, we characterized—at atomic resolution—the stability, electrostatics, water/ions permeability and the dynamic and acoustic properties of the HIV-1 capsid. Our results may provide a new avenue for the development of therapeutic interventions that seek to alter the biophysical properties of the HIV-1 capsid towards the treatment of disease.

## Methods
**Molecular dynamics simulations of the HIV-1 capsid.** In the present study, computer simulations were employed to investigate the stability, electrostatics, ion permeability, acoustics and mechanical properties of the HIV-1 capsid at atomic resolution. In particular, a 1.2 μs MD trajectory of 64,423,983 atoms probed high spatial and temporal resolution characteristics of an empty HIV-1 capsid embedded in its native environment. This level of detail is currently inaccessible to experimental methods alone[26,55].

The HIV-1 capsid composed of 186 CA hexamers and 12 CA pentamers was embedded in a water box with 150 mM sodium chloride, resulting in a simulation box of dimensions 70 nm × 76 nm × 121 nm and a total of 64,423,983 atoms (Fig. 1b), as described in a previous publication[5] and in Supplementary Note 1. In particular, the sequence of HIV-1 subtype B, NL4-3 with the A92E mutation[5], was used for all CA monomers. For the present study, MD simulations of the unrestrained, fully solvated HIV-1 empty capsid were performed using NAMD

2.10 (refs 56,57) for a total of 1.2 µs. MD simulations were performed on the Oak Ridge Leadership Computing Facility (OLCF) TITAN supercomputer (INCITE allocation BIO024) using 3880 GPU accelerated Cray-XK nodes; details regarding the computational challenges of the present simulation are described in Supplementary Note 2. The CHARMM36 (ref. 58) force field was employed with the TIP3P[59] water model at 310 K and 1 atm. Simulations carried out in the present study used the r-RESPA integrator available in NAMD. Long-range electrostatic force calculations employed the particle mesh Ewald method, utilizing a grid spacing of 2.1 Å and eighth-order interpolation with a 1.2 nm cutoff. The simulations employed an integration time step of 2 fs, with nonbonded interactions evaluated every 2 fs and electrostatics updated every 4 fs; all hydrogen bonds were constrained with the SHAKE algorithm.

**Electrostatics and permeability of ions calculations.** Analysis of the capsid electrostatics and ion permeability from the MD trajectories took advantage of OLCF and National Center for Supercomputing Applications (NCSA) Blue Waters high-performance Lustre filesystem, with a stripe count of 16; details regarding the performance of the Lustre filesystem for analysis of biomolecular simulations have been studied by Stone et al.[60] Using OLCF Rhea cluster, electrostatic maps were calculated for the entire system, including protein, water and ions, at intervals of 20 ps over the trajectories, resulting in 20,000 maps. Long-range electrostatic calculations were performed using the particle-mesh-Ewald method in VMD[61,62]. A 3D grid with voxel size of 2 Å was employed for the electrostatics calculations, resulting in $350 \times 343 \times 600$ bins along the $\hat{x}$, $\hat{y}$ and $\hat{z}$ direction, respectively. The 20,000 maps were then averaged using volutil in VMD. Ion occupancies were calculated at 1 Å resolution ($700 \times 646 \times 1,200$ bins) using the volmap plugin in VMD. The occupancy maps were then processed with volutil in VMD to calculate the Gibbs free energy of binding using $\mathcal{W}(\vec{r}) = -k_B T \ln \frac{\rho(\vec{r})}{\rho_0}$, where $\rho(\vec{r})$ is the probability of finding an ion species at a particular site (voxel) as compared with the probability of finding the same ion species in the bulk $\rho_0 = 0.000151$ (explained in detail by Cohen et al.[31]).

**Density and exchange rates of water and ions.** To distinguish between the interior versus exterior of the capsid, a 3D ray-tracing method based on the digital differential analysis algorithm[63] was written in C++ and implemented in VMD. A grid with the same dimensions as the simulation box was created for each frame of the trajectory. To identify the regions of the grid occupied by the protein, the molecular surface of the capsid was calculated using the QuickSurf algorithm in VMD, yielding a continuous surface without holes. Subsequently, for each voxel in the grid, six rays were cast in the $\pm\hat{x}$, $\pm\hat{y}$ and $\pm\hat{z}$ direction, respetively. If any of the rays hit the external wall of the grid, the voxel was considered to be outside of the capsid; conversely, if none of the rays hit an external wall, the voxel was considered to be inside of the capsid. Voxels assigned to the protein were not evaluated using the digital differential analysis algorithm and were classified as protein voxels. Using the classification of each voxel, atoms within a voxel were labelled as belonging to the inside, outside or protein bound. A grid length of 2.0 Å was employed for all calculations. Water density was calculated using the relationship[64], $\rho = \frac{M_w}{0.6022 \times \left(\frac{V}{N}\right)}$, where $M_w$ is the molecular weight of TIP3P water (18.016 g mol$^{-1}$)[64], $V$ is the volume of the region of interest in (Å$^3$) and $N$ is the number of particles in contained the given region.

Exchange rates for water and ions were calculated from the number of molecules that were located on one side of the capsid surface at a reference frame $t_0$ and found on the opposite side of the surface at a later frame $t > t_0$ (ref. 29). The reference frame was selected at different intervals to minimize the effects of molecule recrossing through the boundary of the capsid (Supplementary Fig. 2). Analysis of the densities and exchange rates from the MD trajectories took advantage of NCSA Blue Waters high-performance Lustre filesystem, with a stripe count of 120. Trajectories were analysed in parallel with a load of 12 frames per compute node.

**Determination of acoustic properties.** First, a reference structure for the capsid was determined using the statistical package R[65] as follows. The similarity matrix containing pairwise r.m.s.d. between every time frame of the capsid simulation for the last 400 ns of simulation was generated (using $C_\alpha$s); r.m.s.d. was calculated using the Kabsh algorithm[66]. The parallel implementation of the partition around medoids PAM algorithm available in the SPRINT package[37,67] was utilized to determine the structural medoid of the capsid. Medoids are representative structures observed during the simulation, whose average similarity to all other frames in a trajectory is minimal. Medoids are similar in concept to the mean structure, yet have the advantage of being physical structures, as they are always members of the trajectory. With the medoid as the reference structure, the normalized radial fluctuations for every $C_\alpha$ atom was calculated as $\delta x_i(t) = \frac{\|\vec{x}_i(t)\| - \|\vec{x}_{i,\text{medoid}}\|}{\|\vec{x}_{i,\text{medoid}}\|}$, where $i = 1 \ldots N_{C\alpha}$. Subsequently, periodograms were calculated using R for every fifth $C_\alpha$ (Fig. 5b). A periodogram is the squared magnitude of the discrete Fourier transform and identifies the most dominant frequencies in a time series[36]. Periodograms were then clustered using the Manhattan distance as a similarity score and employing the parallel PAM algorithm in SPRINT[37,67]. The silhouette criteria[37] were evaluated for 2 to

200 clusters, indicating an optimal partioning of the data into four clusters (Supplementary Fig. 6).

**Analysis of collective motions.** The MD trajectory can be written as a matrix $\mathbb{X} = [\mathbf{X}_1 \, \mathbf{X}_2 \cdots \mathbf{X}_n]$, of dimensions $d \times n$, where $d = 3N_{C\alpha} = 451,578$ is the number of cartesian coordinates for all $C_\alpha$s, and $n = 20,000$ is the total number of frames in the trajectory. The mean position of each atom is then given by the vector $\overline{\mathbf{X}} = \frac{1}{n} \sum_{i=1}^{n} \mathbf{X}_i$. Similarly, the covariance matrix of $\mathbb{X}$ is given by $S = \frac{1}{n-1} \sum_{i=1}^{n} (\mathbf{X}_i - \overline{\mathbf{X}})(\mathbf{X}_i - \overline{\mathbf{X}})^{\mathrm{T}}$, with entries, $s_{jk}$ and $s_{jj} = s_j^2$. Therefore, the scaled-centred trajectory can be calculated using[47] $\mathbf{X}_S = S_{\text{diag}}^{-1/2}(\mathbf{X} - \overline{\mathbf{X}})$, where $S_{\text{diag}}$ is the diagonal matrix with entries $s_j^2$ obtained from the covariance matrix $S$. Subsequentely, PCA was performed by calculating the singular value decomposition[47] of the scaled-centred trajectory $\mathbf{X}_S$ by means of the thick-restart Lanczos algorithm[68] in R. In particular, singular value decomposition calculates the $r = \min(d, n)$ eigenvalues $\lambda_{1 \ldots r}$ and eigenvectors/PCs $\eta_i(\lambda)$ of the covariance matrix $S$.

The resulting $\eta_i(\lambda)$ PCs and their associated eigenvalues $\lambda$ were analysed as follows. The cummulative contribution to variance from the first $k$ PCs was calculated using[47] $W^{(k)} = \frac{\sum_{i=1}^{k} \lambda}{\sum_{j=1}^{\min(d,n)} \lambda}$. The participation number of each PC was calculated using[44] $W_\lambda = e^{S_\lambda}$, where $S_\lambda = -\sum p_i \ln(p_i(\lambda))$ is the entropy (information) of the PC. The probabilites $p_i(\lambda)$ are the squared component of the normalized relative displacements for each atom/direction $i$, $p_i(\lambda) = |\eta_i(\lambda)|^2$ (ref. 44). A PC equally distributed over every atom results in a participation number equal to the number of atoms in the system. Conversely, small values of the participation number indicate that the PC is not global[44].

Convergence of the subspace spanned by PCs was evaluated using mutual similarity measurement[40]. Mutual similarity was evaluated using the relationship, $\gamma = \left\langle \frac{\|\mathbf{P}(\mathbf{x})\|^2}{\|\mathbf{x}\|^2} \right\rangle$, where $\langle \cdot \rangle$ denotes an ensemble average over the configurations observed in the trajectory, $\mathbf{x}$ denotes a protein configuration, $\|\cdot\|$ denotes the norm and $\mathbf{P}(\mathbf{x}) = \sum_{i=1}^{m} (\eta_i \cdot \mathbf{x})\eta_i$ denotes the projection to the $m$-dimensional PCA subspace[40]. Importantly, the mutual similariy between PCA subspaces is a number between 0 and 1. To evaluate the convergence of the PCA subspaces, PCs were calculated for trajectory fragments of different size, ranging from 50 to 200 ns. Subsequently, mutual similarities, using 10 to 1,000 PCs, were calculated between fragments using R and plotted as a function of fragment size (Supplementary Fig. 7).

**Data availability.** The data that support the findings of this study are available from the corresponding author on request.

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

## Acknowledgements

We thank Jodi Hadden for critical reading of the manuscript and editorial support. We are also grateful to Jim Phillips and John Stone for enlightening conversations; Chris Aiken, Angela M. Gronenborn, Tatyana Polenova, Peijun Zhang and all members of the Pittsburgh Center for HIV Protein Interactions for insightful discussions. We acknowledge funding by the National Institutes of Health Grants 9P41GM104601. This research is part of the Blue Waters sustained-petascale computing project supported by NSF awards OCI-0725070 and ACI-1238993, the state of Illinois and the 'Computational Microscope' NSF PRAC award ACI-1440026. An award of computer time was provided by the Innovative and Novel Computational Impact on Theory and Experiment program (INCITE award BIO024). This research also used resources from the Oak Ridge Leadership Computing Facility at Oak Ridge National Laboratory that is supported by the Office of Science of the Department of Energy under Contract DE-AC05-00OR22725.

## Author contributions

The present study was conceived by J.R.P. and K.S. Computational methodology was developed by J.R.P and K.S. Molecular dynamics simulation and structure building tools were developed by K.S. Simulations were designed by J.R.P. J.R.P. constructed structural models and performed MD simulations. J.R.P. and K.S. designed analysis framework for large-scale simulations. J.R.P. analysed the MD simulations. J.R.P. and K.S. wrote the paper.

## Additional information

**Competing interests:** The authors declare no competing financial interests.

**Publisher's note**: 

