## [Peer Review File · Nature Communications]

Reviewers' comments:

Reviewer #1 (Remarks to the Author):

The manuscript reports on the cutting edge simulations of an entire HIV capsid in explicit water. The results are statistically valid and the conclusions are convincing, they represent significant new contribution interesting both for specialised virusology and general audience. Therefore, I support publishing the manuscript in Nature Communications after addressing the following issues.

- There is no description of how the initial structure for the simulation has been prepared. This is a very important point as for this large system the structure is unlikely to evolve far away from the initial configuration. Thus, the results and conclusions depend on the correct initial structure. Perhaps the description is in the cited publications, but it is important to have it included in this manuscript as well.
- The conclusion that the structure is stable is based on the RMSD analysis of the non-flexible parts of the proteins, Fig. 2a. However, it is clear that the curve does not have a plateau signifying that the deviation from the initial structure continues to grow all the time even at the end of the simulated period. The overall structure is most likely stable, as the authors claim, because the cross-section area and the height of the capsid do have plateaus, Fig. 2c,d. However, the smaller scale details are apparently not equilibrated. The authors should investigate what processes cause this change of RMSD and show that they do not affect the stability of the capsid as a whole.
- It is also interesting to note the long time scale fluctuations of the cross-section area and the height, Fig. 2c,d. Perhaps they are related to the structural fluctuations described later in the text. It would be good to see authors' comments on this.
- The analysis of electrostatics lacks the time component: only an averaged distribution is reported. It will be interesting to see the change of the electrostatics during simulation. What scales are these processes? Are they related to the structural fluctuations?
- Related to the previous point is the analysis of how the distribution of ions changes in time. In particular, how it was equilibrated from the initial distribution (which is also not described). As the ions move slowly, it is not clear if the equilibrium distribution has been obtained in the simulation.
- The time evolution description is also necessary for the reported acoustic properties: how the identified clusters change in time? The simulated data seems to be enough for performing at least some rough analysis of time variations.
- In Fig. 6c,d, S2 the arrows are not visible that makes it impossible to see collective motion of the atoms described in the text. Perhaps some colour indication would help here?

Reviewer #2 (Remarks to the Author):

Review report to
"Physical properties of HIV-1 capsid from all-atom molecular dynamics simulations", by J.R. Perilla and K. Schulten

Summary of review:

The manuscript comprehensively discusses the chemical-physical properties of entire HIV-1 capsids in NaCl solution, without RNA, by running large-scale all-atomistic molecular dynamics (MD) calculations, at microsecond time-scale, taking advantage of one of the biggest modern supercomputer systems. Stability of the capsid, its ion permeability, electrostatics around the capsid, and acoustic and mechanical properties of capsid have been investigated within the limitation of simulation time (1.2 microseconds). Viewpoints of the analyses are interesting, where the results could contribute to a deeper understanding of chemical-physical properties of the empty HIV-1 capsid. However, I judged that some of the results do not reach the level of accuracy to make clear conclusions on the HIV-1 capsid's general chemical-physical properties. Further, some of the analyses stop at an observation: There is no subsequent discussion on the cause of the observations, accompanied by detailed analysis to explain their mechanism on a molecular level. Therefore, I recommend the authors to resubmit the manuscript with thorough corrections and with further detailed analyses according to the points listed below.

Review comments:

In the introduction section,

1. At the end of introduction, it should be summarized how the results of analyses executed in this manuscript could contribute to understanding of biological function of HIV-1 capsids, accompanied by experimental implications if possible, which may be main concern of manuscript readers. At present, one of them is placed at intervals, which makes it difficult to follow.
2. The HIV capsid handled in MD calculations is nothing more than a model of natural HIV-1 capsids. This point should be emphasized in an introduction section referring to their previous work [5] to avoid first time readers from misunderstanding as if it were the naturally occurring HIV-1 capsid. The lack of RNA inside capsid should also be emphasized. For example, in Fig.1 caption, a phrase "The HIV-1 capsid model[5] without genome" instead of "The HIV-1 capsid" should be used to avoid misleading the readers.

In 'Stability of the HIV-1 capsid' subsection,

3. Have the authors checked for the stability of the capsid as a whole? Do some changes take place during the initial 400 ns simulation, where they observed shrinking of the capsid. For example, the RMSDs for the centers of mass of pentamers and hexamers from the starting structure will give us an information on how much a whole structure of capsid deviates from initial one. Further, whether the whole structure reaches to equilibrium state or not after 1.2 microsecond calculation should be discussed based on mass center RMSDs.
4. Surprisingly, the authors are not concerned about what causes the shrinkage of the capsid. What is a driving force which caused the shrinkage? After 400 ns simulation, why is the shrinkage subsided? The authors must answer these questions which article readers will have. In my guess, equilibration of solvent distribution (water and ions) by crossing the capsid may closely relate to the shrinkage.

In 'Electrostatics and permeability to ions of the HIV capsid' subsection,

5. The obtained electrostatic potential map indicates that the regions inside and outside of the capsid are at the same electrostatic potential. The equality of electrostatic potential between inside and outside capsid seems to be intriguing, because the capsid has highly negative charge (-3528e). Apparently, the capsid itself produces spatially non-uniform electrostatic potential distribution in vacuum. It implies that solvent would produce electrostatic potential distribution around the capsid to counteract the non-uniform distribution produced by the capsid. Can the authors show an evidence for this mechanism?
6. It is surprising to observe that the highly negatively charged capsid (-3528e) has a higher occupancy of negatively charged chloride ions than that of positively charged sodium ions. Is it possible to explain to why this discrepancy occurs?

7. The authors only discuss about local distribution of ions around CA hexamers and pentamers. However, most readers would be interested to know what were initial distributions of water and ions around the capsid and what their equilibrated distributions are, after 400 ns of simulation. Further, where and how fast solvent exchanges across the capsid is hot topic attracting interests in the field of virology [as in the recent paper J. Chem. Phys., 141, 165101 (2014), for poliovirus empty capsids]. Thus, I strongly request the authors to add discussions with (supplementary) figures to answer these questions about water and ions distributions.

8. Obviously, a calculation of free energy difference ΔG based on a probability distribution assumes that a system is in a thermally equilibrated state. However, the authors seem to not consider this assumption at all. To show the system does satisfy this assumption, it is necessary to show an equivalence of the chemical potential for water and ions between capsid inside and outside after 400 ns simulation.

9. The binding energy comparison of sodium in HIV-1 capsid, which is a structural protein, to thrombin, which is an enzyme, is not appropriate. Instead, the authors should cite examples of ion binding to structural proteins.

10. Does lower binding of chloride ions to hexamer than pentamer indicate that the ions diffuse faster through the hexamer channels? Can the authors explain what causes the lower binding energies?

11. In this subsection, the authors have omitted the in vivo environment of the capsid interior, although there is a strong negatively charged nucleic acid in nature. It is desirable to add comments about possible changes on the results for electrostatic potential maps around the capsid and ion binding energy (i.e. distribution of ions) in the presence of RNA and other proteins in the HIV-1 capsid.

In 'Acoustic properties of the HIV-1 capsid' subsection,

12. Why did the authors calculate the acoustic properties of the capsid after 300 ns (noted in Fig5's caption, is it misspelled?) during which they had observed shrinkage of the capsid. Probably after 400 ns when the capsid structure may reach a stationary point, the authors could have calculated whether the oscillatory behavior persists or changes. The best way to answer my question is that interval averages (two 400 ns intervals during 400 ns - 1200 ns, for example) of acoustic properties are shown in supplementary material, and discuss their equivalences between interval averages with statistical error analysis.

13. What is the biological implication of the oscillation observed?

14. Can the authors comment on why they observed higher fundamental frequency for certain clusters? Is it due to the structural asymmetry? Do the frequencies imply direction of ion/molecule diffusion?

In 'Mechanical models of HIV-1 capsid' subsection,

15. To discuss a general science relating to collective atomic motions in HIV-1 capsid, which is the aim of this manuscript, the principal component analysis (PCA) should be done on the trajectories after reaching equilibrium state. However, the authors do not check this point critically. Similar to Q.12 listed above, independent PCA analyses should be performed on divided trajectories with the same time length, then showing equivalency of the PCA results between them within a statistical error. Without such a validation, readers cannot judge whether the belt shown in Fig.6d is stationary or not. Critically reading, the belt may be a cause of shrinkage of the capsid, and it would disappear when the capsid reaches mechanical equilibrium.

16. In the principal component analysis, a two dimensional projection of PC1 vs PC2 is often shown to discuss correlation between main two principal components. Is it possible to discuss correlation of PC1 and PC2 to the stability of capsid?

17. What do the different motions in the viral capsid imply from a biological point of view?

In 'Discussion' section,

18. As described above, evidences for conclusion are weak at present. Most of discussions should be supported by more detailed analysis to explain what element caused these observations. How spatially uniform electrostatic potential is realized, whether water/ions distributions are equilibrated, whether acoustic and mechanical properties are stationary, should be validated before making conclusions on general properties of HIV-1 capsids.

19. Recently, other group published an article on HIV-1 capsid self-assembly based on coarse-grained simulation [Nature Commn., 7, 11568 (2016).]. Is it possible to make comparative discussion with it? If not, at least, it is necessary to cite it in an introduction section to open discussion to readers.

In 'Methods' section,

20. Subsection named "Computational challenges" seems to be out of scope for most readers of Nature Communications. It should be separately submitted to journals in the field of computer science or data science.

Errors on editing:

1. Spelling and grammar mistakes

In abstract, at first, "hummn" should be "human"

Line 24 on page 5: Title of subsection 'Electrostatics and permeability to ions of ...' should be 'Electrostatics and ion permeability of ...'

Lines 6-7 on page 4: grammar needs correction

Fig.2 caption: "(A) ... moving average with a window size of 10 ns is shown in red" may be "shown in blue".

2. Figure notation

All figures: Small letters (a, b, c, ...) are used to represent panel numbers, whereas large letters (A, B, C, ...) used in their captions. Unify the notation following to guide to authors. Figures except for Fig.2 lack descriptions of definition of error bars and sample (or interval) number.

Fig.1A: Which is the CTD and NTD? It needs to be highlighted according to the text.

Fig.6D: The "belt" has not been adequately represented. Probably, highlighting the particular motions might be useful for the readers.

Reviewer #3 (Remarks to the Author):

The authors present an all-atom MD simulation of the mature HIV capsid, over 1.2 microsec of time. This is an impressive feat. The work is novel and important. The manuscript is well written and well referenced overall, and it will be widely cited by other structural virologists. Overall, the quality looks excellent. My only reservation is the biological significance. The authors do put a very conventional statement at the end of the discussion, that this detailed understanding of the HIV capsid should underpin development of anti-HIV pharmaceutical compound in the future. But this statement is pretty futuristic. This comment notwithstanding, I do think that the manuscript deserves to be published because of its novelty. Few other labs are capable of this kind of careful computational work on virus structure.

Minor comments:

1. Fig 1 legend: "fullerene" is not a noun. Presumably the authors mean "fullerene structures" or something like that. Also here: "neutralizing" is misspelled.

2. page 4, line 6. "elucidating" is misspelled.

3. page 5, line 4: presumably the authors want to use the past tense "probed" here.

4. page 5, line 6: "which" instead of "that".

5. page 5, lines 12, 14, and 15: use uniform tense: "...hexamers and pentamers ...showed..."; "...respectively showed..."-since you used "was" in line 15.

6. Page 7, line 6: "intriguingly" is misspelled.

7. page 16, lines 2,3: "finally, our observations of ion-specific binding sites...high salt concentrations..." I don't follow this logic. High ionic strength to me minimizes electrostatic repulsion, but doesn't require any "specific" ion-binding site. Please clarify.

Response to reviewers' critiques and comments:

(Reviewers' comments are in black; **Authors' comments are in blue.**)

Reviewer #1 (Remarks to the Author):

The manuscript reports on the cutting edge simulations of an entire HIV capsid in explicit water. The results are statistically valid and the conclusions are convincing, they represent significant new contribution interesting both for specialised virusology and general audience. Therefore, I support publishing the manuscript in Nature Communications after addressing the following issues.

We thank the reviewer for taking the time to review our manuscript and provide a critical assessment of our work. We thank the reviewer for her/his supportive comments.

- There is no description of how the initial structure for the simulation has been prepared. This is a very important point as for this large system the structure is unlikely to evolve far away from the initial configuration. Thus, the results and conclusions depend on the correct initial structure. Perhaps the description is in the cited publications, but it is important to have it included in this manuscript as well.

Although we described the preparation of the system studied in the present manuscript in a previous publication (Zhao et al. Nature 2013), we have incorporated a new section in the supplemental information which describes in detail the construction of the system and the simulation box.

- The conclusion that the structure is stable is based on the RMSD analysis of the non-flexible parts of the proteins, Fig. 2a. However, the curve does not have a plateau signifying that the deviation from the initial structure continues to grow all the time even at the end of the simulated period. The overall structure is most likely stable, as the authors claim, because the cross-section area and the height of the capsid do have plateaus, Fig. 2c,d. However, the smaller scale details are apparently not equilibrated. The authors should investigate what processes cause this change of RMSD and show that they do not affect the stability of the capsid as a whole.

The reviewer raises an important point regarding the stability of the entire capsid in terms of its structural integrity. To show that the structural integrity of the entire capsid is maintained over the entire trajectory, we have included supplemental figures showing the coarse-grained RMSD evolution (Figure S1). In addition, we have added a comment regarding the stability of the entire capsid to the results sections. We have also included a movie showing a CG-representation of the capsid to illustrate its stability as a whole (Movie M1).

- It is also interesting to note the long time scale fluctuations of the cross-section area and the height, Fig. 2c,d. Perhaps they are related to the structural fluctuations described later in the text. It would be good to see authors' comments on this.

We thank the reviewer for bringing out an interesting point. Certainly, the fluctuations in height and width are also mapped by our acoustic analysis, as described in the acoustic section of the manuscript.

- The analysis of electrostatics lacks the time component: only an averaged distribution is reported. It will be interesting to see the change of the electrostatics during simulation. What scales are these processes? Are they related to the structural fluctuations?

The reviewer suggests a clever analysis. We have presented a series of electrostatic potentials over the course of the simulation corresponding to shorter averages of the electrostatics produced by the capsid. The results do not show straightforward differences between snapshots of the electrostatics maps, but support our observation that the interior and exterior of the capsid exhibit the same electrostatic potential.

- Related to the previous point is the analysis of how the distribution of ions changes in time. In particular, how it was equilibrated from the initial distribution (which is also not described). As the ions move slowly, it is not clear if the equilibrium distribution has been obtained in the simulation.

We thank the reviewer for bringing to our attention such a critical point (also raised by reviewer #2). We evaluated the change of ion distributions during the simulation and identified that thermal equilibrium for both ion species was reached; both points are illustrated in Figures S2C and Figure S2D. Similarly, we show the distribution of ions on the entire capsid, revealing a binding pattern beyond the one previously reported by us during the initial submission (Figure 4).

- The time evolution description is also necessary for the reported acoustic properties: how the identified clusters change in time? The simulated data seems to be enough for performing at least some rough analysis of time variations.

We thank the reviewer for such clever observation regarding the statistical robustness of our acoustic analysis. To address this potential pitfall, we have performed a systematic analysis based on subsamples of the trajectory. The analysis reveals that clusters are still present in smaller portions of the trajectory, nonetheless the mapping of frequencies to structure seem to correspond to different regions of the capsid for each subsample. The differences in frequency-to-structure mapping suggests that the waves we observe are not stationary. We therefore conclude that the waves observed behave like capillary waves common in several types of membranes. We mention this point in the acoustic section of the manuscript and present our statistical analysis in the supplemental material (Figures S4 and S5).

- In Fig. 6c,d, S2 the arrows are not visible that makes it impossible to see collective motion of the atoms described in the text. Perhaps some colour indication would help here?

We thank the reviewer for raising this visualization concern. We have improved the figure for visual clarity.

Reviewer #2 (Remarks to the Author):

Review report to
"Physical properties of HIV-1 capsid from all-atom molecular dynamics simulations", by J.R. Perilla
and K. Schulten

Summary of review:

The manuscript comprehensively discusses the chemical-physical properties of entire HIV-1 capsids in NaCl solution, without RNA, by running large-scale all-atomistic molecular dynamics (MD) calculations, at microsecond time-scale, taking advantage of one of the biggest modern supercomputer systems. Stability of the capsid, its ion permeability, electrostatics around the capsid, and acoustic and mechanical properties of capsid have been investigated within the limitation of simulation time (1.2 microseconds). Viewpoints of the analyses are interesting, where the results could contribute to a deeper understanding of chemical-physical properties of the empty HIV-1 capsid. However, I judged that some of the results do not reach the level of accuracy to make clear conclusions on the HIV-1 capsid's general chemical-physical properties. Further, some of the analyses stop at an observation: There is no subsequent discussion on the cause of the observations, accompanied by detailed analysis to explain their mechanism on a molecular level. Therefore, I recommend the authors to resubmit the manuscript with thorough corrections and with further detailed analyses according to the points listed below.

We thank the reviewer for taking the time to read our manuscript and provide a critical assessment of our work. Below we respond to each of the concerns raised by the reviewer.

Review comments:

In the introduction section,

1. At the end of introduction, it should be summarized how the results of analyses executed in this manuscript could contribute to understanding of biological function of HIV-1 capsids, accompanied by experimental implications if possible, which may be main concern of manuscript readers. At present, one of them is placed at intervals, which makes it difficult to follow.

We appreciate the comment from the reviewer. We have incorporated the suggested changes to the introduction together with relevant references (page 4, line 5).

2. The HIV capsid handled in MD calculations is nothing more than a model of natural HIV-1 capsids. This point should be emphasized in an introduction section referring to their previous work [5] to avoid first time readers from misunderstanding as if it were the naturally occurring HIV-1 capsid. The lack of RNA inside capsid should also be emphasized. For example, in Fig.1 caption, a phrase "The HIV-1 capsid model[5] without genome" instead of "The HIV-1 capsid" should be used to avoid misleading the readers.

We have changed the wording to make sure that our readers are not confused, we refer to the empty virus cores as virus-like-particles (VLPs). We thank the reviewer for pointing out this possible semantic pitfall.

In 'Stability of the HIV-1 capsid' subsection,

3. Have the authors checked for the stability of the capsid as a whole? Do some changes take place during the initial 400 ns simulation, where they observed shrinking of the capsid. For example, the RMSDs for the centers of mass of pentamers and hexamers from the starting structure will give us an information on how much a whole structure of capsid deviates from initial one. Further, whether the whole structure reaches to equilibrium state or not after 1.2 microsecond calculation should be discussed based on mass center RMSDs.

A similar point was raised by reviewer #1, that is, the stability of the entire capsid in terms of its structural integrity. To show that the structural integrity of the entire capsid is maintained over the entire trajectory we have included a supplemental figure (Figure S1). In addition, we have added a comment regarding the stability of the entire capsid to the results sections. We have also included a movie showing a CG-representation of the capsid to illustrate its stability (Movie M1).

4. Surprisingly, the authors are not concerned about what causes the shrinkage of the capsid. What is a driving force which caused the shrinkage? After 400 ns simulation, why is the shrinkage subsided? The authors must answer these questions which article readers will have. In my guess, equilibration of solvent distribution (water and ions) by crossing the capsid may closely relate to the shrinkage.

We thank the reviewer for the astute observation. Indeed, the water density seems to be a key player in the shrinking of the capsid. We have presented the equilibration of water density in Figure S1. Similarly, we have performed extended analysis on the translocation for both ion species.

In 'Electrostatics and permeability to ions of the HIV capsid' subsection,

5. The obtained electrostatic potential map indicates that the regions inside and outside of the capsid are at the same electrostatic potential. The equality of electrostatic potential between inside and outside capsid seems to be intriguing, because the capsid has highly negative charge (-3528e). Apparently, the capsid itself produces spatially non-uniform electrostatic potential distribution in vacuum. It implies that solvent would produce electrostatic potential distribution around the capsid to counteract the non-uniform distribution produced by the capsid. Can the authors show an evidence for this mechanism?

We have included a new panel in Figure 3, which shows the distribution of ions in the context of the entire capsid; the distribution of ions counter-acts the electrostatic potential distribution produced by the empty capsid.

6. It is surprising to observe that the highly negatively charged capsid (-3528e) has a higher occupancy of negatively charged chloride ions than that of positively charged sodium ions. Is it possible to explain to why this discrepancy occurs?

Careful inspection of Figure 4c shows that the binding sites of sodium are generally on the exterior of the capsid; furthermore, bound sodium seem to be more exposed to the bulk solvent, compared to the central pores where chloride binds (Figure 4a&b). We have made a comment about this apparent discrepancy in the body of the manuscript (Page 9, line 5).

7. The authors only discuss about local distribution of ions around CA hexamers and pentamers. However, most readers would be interested to know what were initial distributions of water and ions around the capsid and what their equilibrated distributions are, after 400 ns of simulation. Further, where and how fast solvent exchanges across the capsid is hot topic attracting interests in the field of virology [as in the recent paper *J. Chem. Phys.*, 141, 165101 (2014), for poliovirus empty capsids]. Thus, I strongly request the authors to add discussions with (supplementary) figures to answer these questions about water and ions distributions.

We have included a new discussion and supplemental figures to address the transfer rates of water and ions (Figure S2). In addition, we illustrate the location of these ion channels on the surface of the capsid (Figure 4C). We employ some of the methodologies developed in *J. Chem. Phys.*, 141, 165101 (2014); and explain them in the methods section. We thank the reviewer for bringing this analysis into our manuscript.

8. Obviously, a calculation of free energy difference ΔG based on a probability distribution assumes that a system is in a thermally equilibrated state. However, the authors seem to not consider this assumption at all. To show the system does satisfy this assumption, it is necessary to show an equivalence of the chemical potential for water and ions between capsid inside and outside after 400 ns simulation.

We apologize for overlooking such an important condition for calculating free energy differences. As illustrated in Figure S1, the exchange rate for each ionic species is in thermal equilibrium; therefore, there is equivalence between the chemical potential for water and ions between the interior and exterior of the capsid. We thank the reviewer for bringing this key element to our manuscript.

9. The binding energy comparison of sodium in HIV-1 capsid, which is a structural protein, to thrombin, which is an enzyme, is not appropriate. Instead, the authors should cite examples of ion binding to structural proteins.

We have included references to other structural proteins that are known to bind ions: the glutamate receptor [*PNAS*, 2010 vol. 107 no. 31 13912-13917] and the sodium symporter [*Molecular Cell*, 2008, Volume 30, Pages 667-677].

10. Does lower binding of chloride ions to hexamer than pentamer indicate that the ions diffuse

faster through the hexamer channels? Can the authors explain what causes the lower binding energies?

We thank the reviewer for pointing out this critical point. The diameter of the pores in the pentamers is smaller than of the hexamers. This geometrical characteristic of the pore seems to have a direct impact on the transfer rate. We have made comments regarding this point in the manuscript (page 10, line 14).

11. In this subsection, the authors have omitted the in vivo environment of the capsid interior, although there is a strong negatively charged nucleic acid innature. It is desirable to add comments about possible changes on the results for electrostatic potential maps around the capsid and ion binding energy (i.e. distribution of ions) in the presence of RNA and other proteins in the HIV-1 capsid.

We thank the reviewer for the astute observation. We have included a comment on the possible changes resulting from the presence of RNA and other proteins inside of the capsid core to the discussion section (page 17, line 16).

In 'Acoustic properties of the HIV-1 capsid' subsection,

12. Why did the authors calculate the acoustic properties of the capsid after 300 ns (noted in Fig5's caption, is it misspelled?) during which they had observed shrinkage of the capsid. Probably after 400 ns when the capsid structure may reach a stationary point, the authors could have calculated whether the oscillatory behavior persists or changes. The best way to answer my question is that interval averages (two 400 ns intervals during 400 ns - 1200 ns, for example) of acoustic properties are shown in supplementary material, and discuss their equivalences between interval averages with statistical error analysis.

We thank the reviewer for pointing out this potential issue in our writing; we note however that the acoustic analysis was performed in the last 400 ns of the simulation. However, Figure5 panel A illustrates the change on the surface after 400 ns of simulation, between 800ns and 1200ns. We have made sure that is clear to the reader that the analysis was performed over the last 400 ns of simulation. In addition, to establish the robustness of our calculations we have performed acoustic analysis on halved and quartered trajectories and evaluated the convergence of our results. Therefore, we have included a new discussion in the text of the manuscript and included new supplemental figures to illustrate the convergence of our acoustic analysis (Figure S5 and Figure S6).

13. What is the biological implication of the oscillation observed?

We speculate that these oscillations play an important role during nuclear import, when the virus capsid interacts with the nuclear pore complex. We have generated a model of the capsid interacting with the nuclear pore complex based on the experimental density of the nuclear pore complex and our capsid model, and added a figure of the system to the supplemental material (Figure S10). In addition, we have incorporated our view to the discussion section of the manuscript (page 16, line 20).

14. Can the authors comment on why they observed higher fundamental frequency for certain clusters? Is it due to the structural asymmetry? Do the frequencies imply direction of ion/molecule diffusion?

It seems that the fundamental frequencies are not related to structural asymmetry (Figure S6). The second question raised by the reviewers is rather provocative; nonetheless, we also believe that the question would be better addressed in a future publication and leave the comment as an open question in the manuscript (page 13, line 1).

In 'Mechanical models of HIV-1 capsid' subsection,

15. To discuss a general science relating to collective atomic motions in HIV-1 capsid, which is the aim of this manuscript, the principal component analysis (PCA) should be done on the trajectories after reaching equilibrium state. However, the authors do not check this point critically. Similar to Q.12 listed above, independent PCA analyses should be performed on divided trajectories with the same time length, then showing equivalency of the PCA results between them within a statistical error. Without such a validation, readers cannot judge whether the belt shown in Fig.6d is stationary or not. Critically reading, the belt may be a cause of shrinkage of the capsid, and it would disappear when the capsid reaches mechanical equilibrium.

We present the convergence of our PCA analysis making use of the concept of mutual similarity of principal components between fragments of the trajectory with the same length. The results from such analysis show that the subspace spanned by the principal components is converged for the last 400 ns of simulation (Figure S7). We thank the reviewer for bringing this analysis into our manuscript.

16. In the principal component analysis, a two dimensional projection of PC1 vs PC2 is often shown to discuss correlation between main two principal components. Is it possible to discuss correlation of PC1 and PC2 to the stability of capsid?

We present the projection of the trajectory into the first two principal components (Figure S8). We also added a discussion regarding the stability of the capsid and the two-dimensional projection of the trajectory to the manuscript.

17. What do the different motions in the viral capsid imply from a biological point of view?

The motions of the capsid could have several biological implications. One of them is related to the uncoating mechanism by which the capsid breaks apart, releasing its genetic material. Recently, it has been shown that uncoating of the HIV-1 capsid mechanically initiates exhibiting changes in the internal pressure of the capsid which results in its rupture on the narrow end [Journal of Virology, 10.1128/JVI.00289-17

2017]. We have made comments regarding this connection to the biology of HIV-1 in the text (page 17, line 22).

In 'Discussion' section,

18. As described above, evidences for conclusion are weak at present. Most of discussions should be supported by more detailed analysis to explain what element caused these observations. How spatially uniform electrostatic potential is realized, whether water/ions distributions are equilibrated, whether acoustic and mechanical properties are stationary, should be validated before making conclusions on general properties of HIV-1 capsids.

We appreciate the feedback provided by the reviewer. As mentioned before, we have addressed every point raised by the reviewer: spatially uniform electrostatics, equilibration of water and ions distributions, as well as the statistical significance of the acoustic and mechanical properties. We are grateful to the reviewer for all her/his suggestions.

19. Recently, other group published an article on HIV-1 capsid self-assembly based on coarse-grained simulation [Nature Commn., 7, 11568 (2016)]. Is it possible to make comparative discussion with it? If not, at least, it is necessary to cite it in an introduction section to open discussion to readers.

Although our work doesn't explore self-assembly of the capsid, or virus like particle. An immediate connection between the work presented in [Nature Comm., 7, 11568 (2016).] and our work is apparent: a global arrangement of the capsid happens first, followed by local re-arrangements of the capsid i.e., global shrinking followed by a stable structure. We have made a note about the link between the two manuscripts in the discussion section of our manuscript (page 16, line 8).

In 'Methods' section,

20. Subsection named "Computational challenges" seems to be out of scope for most readers of Nature Communications. It should be separately submitted to journals in the field of computer science or data science.

We have moved the "Computational challenges" section from the methods section to the supplemental information. We expect to submit a manuscript to specialized journals in the future.

Errors on editing:

1. Spelling and grammar mistakes

In abstract, at first, "hummn" should be "human"

Line 24 on page 5: Title of subsection 'Electrostatics and permeability to ions of ...' should be 'Electrostatics and ion permeability of ...'

Lines 6-7 on page 4: grammar needs correction

Fig.2 caption: "(A) ... moving average with a window size of 10 ns is shown in red" may be be "shown in blue".

2. Figure notation

All figures: Small letters (a, b, c, ...) are used to represent panel numbers, whereas large letters (A, B, C, ...) used in their captions. Unify the notation following to guide to authors. Figures except for Fig.2 lack descriptions of definition of error bars and sample (or interval) number.

Fig.1A: Which is the CTD and NTD? It needs to be highlighted according to the text.

Fig.6D: The "belt" has not been adequately represented. Probably, highlighting the particular motions might be useful for the readers.

We have addressed all the editing issues raised by all reviewers.

Reviewer #3 (Remarks to the Author):

The authors present an all-atom MD simulation of the mature HIV capsid, over 1.2 microsec of time. This is an impressive feat. The work is novel and important. The manuscript is well written and well referenced overall, and it will be widely cited by other structural virologists. Overall, the quality looks excellent. My only reservation is the biological significance. The authors do put and very conventional statement at the end of the discussion, that this detailed understanding of the HIV capsid should underpin development of anti-HIV pharmaceutical compound in the future. But this statement is pretty futuristic. This comment notwithstanding, I do think that the manuscript deserves to be published because of its novelty. Few other labs are capable of this kind of careful computational work on virus structure.

Minor comments:

1. Fig 1 legend: "fullerene" is not a noun. Presumably the authors mean "fullerene structures" or something like that. Also here: "neutralizing" is misspelled.

2. page 4, line 6. "elucidating" is misspelled.

3. page 5, line 4: presumably the authors want to use the past tense "probed" here.

4. page 5, line 6: "which" instead of "that".

5. page 5, lines 12, 14, and 15: use uniform tense: "...hexamers and pentamers ...showed..."; ..."respectively showed..."-since you used "was" in line 15.

6. Page 7, line 6: "intriguingly" is misspelled.

7. page 16, lines 2,3: "finally, our observations of ion-specific binding sites...high salt concentrations..." I don't follow this logic. High ionic strength to me minimizes electrostatic repulsion, but doesn't require any "specific" ion-binding site. Please clarify.

We thank the reviewer for taking the time to read our manuscript and provide a critical assessment of our work. We thank the reviewer for her/his positive comments. The issues raised by the reviewers have been addressed, as shown in the attached document. We want to point out that in the past year some publications have experimentally made observations in line with our computational predictions. For instance, it has been shown that blocking of the central pores in HIV-1 capsid cores reduces reverse transcription [Jacques et al. Nature 2016], and that uncoating is mechanically initiated and results in rupture of the narrow end of the capsid [Rankovic et al. Journal of Virology 2017]).

REVIEWERS' COMMENTS:

Reviewer #1 (Remarks to the Author):

The authors have done significant amount of additional analysis. They have also added substantial discussions and conclusions to the text, answering all my concerns. I am satisfied with the answers. I believe the manuscript can be published in the present form.

Reviewer #2 (Remarks to the Author):

The authors have revised the manuscript with great effort. The data additionally included in the main text and supplementary material, together with related discussions, provide the evidence of their conclusions and the base for future discussions. The quality of the revised manuscript has reached to a quite high level. Therefore, I recommend this article to be published in Nature Communications.

Minor comments:

(1) There is discrepancy between labels of figure panel used in the main text and that in the figure.

For example, "Figure 1A" (capital letter A) is used in the main text, while 'a' (small letter a) is used in the left panel in Figure 1.

These should be corrected, if necessary.

(2) "Mhz" (mega hertz) should be "MHz".

Reviewer #3 (Remarks to the Author):

The authors have satisfactorily addressed the various points in the original critiques. In my opinion the manuscript is an excellent contribution to understanding of the HIV lattice. I have no concerns and look forward to seeing this paper published.